# Clipping Noise Compensation with Neural Networks in OFDM Systems

**Tzu-Hsien Sang ***  **and You-Cheng Xu**

Institute of Electronics, National Chiao Tung University, Hsin-Chu 300, Taiwan; knight519673@gmail.com
* Correspondence: tzuhsien54120@faculty.nctu.edu.tw; Tel.: +886-573763-54120

**Abstract:** The application of deep learning (DL) to solve physical layer issues has emerged as a prominent topic. In this paper, the mitigation of clipping effects for orthogonal frequency division multiplexing (OFDM) systems with the help of a Neural Network (NN) is investigated. Unlike conventional clipping recovery algorithms, which involve costly iterative procedures, the DL-based method learns to directly reconstruct the clipped part of the signal while the unclipped part is protected. Furthermore, an interpretation of the learned weight matrices of the neural network is presented. It is observed that parts of the network, in effect, implement transformations very similar to the (Inverse) Discrete Fourier Transform (DFT/IDFT) to provide information in both the time and frequency domains. The simulation results show that the proposed method outperforms existing algorithms for recovering clipped OFDM signals in terms of both mean square error (MSE) and Bit Error Rate (BER).

**Keywords:** OFDM; clipping; signal recovery; deep learning; interpretation of deep learning

## 1. Introduction

Orthogonal frequency division multiplexing (OFDM) is a popular multi-carrier modulation scheme due to the advantages of its high data rates, high bandwidth efficiency and robustness against multi-path fading [1]. However, OFDM has its issues. One of its main problems is its high peak power to average power ratio (PAPR), i.e., its signal exhibits large envelope fluctuations [2]. This property makes the power amplifier (PA) operate beyond its comfortable working range, which results in low power efficiency and induces nonlinear distortion to the transmitted signal. Consequently, it is highly desirable for OFDM systems to reduce PAPR to avoid these adversarial effects. Many PAPR reduction algorithms have been proposed; for instance, partial transmit sequences, selective mapping, and tone reservation [3–5] are among the most prominent, but usually high computational effort is required. Typically, a reduction of at least 5 dB is expected from these reduction techniques when the probability that the original peak exceeds a preset level is set to $10^{-6}$ [6]. Here, we consider the simplest PAPR reduction technique, namely digital clipping, which is easily implemented in the transmitter. However, digital clipping causes the problems of in-band distortion and out-of-band leakage; therefore, a potent clipping compensation scheme at the receiver is needed to work with digital clipping to maintain the overall system performance [6–8].

Deep learning (DL) has the potential to be a powerful tool for solving physical layer issues [9], such as modulation recognition, channel estimation and detection [10–12], the decoding of linear codes [13], and estimating error rates from system parameters [14]. As far as clipping is concerned, it is shown in [10] that DL-based signal detection is quite resistant to clipping noise. However, as clipping becomes severe, performance degrades dramatically and a dedicated compensation scheme is needed. Several approaches exist to compensate clipping effects at the receiver or transmitter side. It is possible to filter clipping effects as impulse noises if their occurrence is limited [15]. A transmitter-side technique

proves quite effective at lowering the BER floor with a clipping–filtering procedure [16]. Dedicated receiver-side algorithms such as those in [17,18], though, offer a supreme performance and rely on costly computation and high-latency iterative procedures to "reconstruct" clipped signal peaks. It is desirable to have a clipping noise compensation scheme, such that a superior performance can be obtained with a low computational cost and low latency. Algorithms based on Neural Networks (NNs) with simple feedforward operations look promising to fulfill these requirements.

The working mechanism inside a neural network is known to be very opaque and resistant to explanations. Nonetheless, in some cases, it is possible to understand and interpret the neural network in a much clearer way; for instance, it has been shown that a network architecture can be obtained by unfolding an iterative optimization algorithm [19]. We believe that the compensation of clipping noise may offer such a scenario, wherein the weight matrices of the NN yield a meaningful interpretation. The iterative nature of clipping noise compensation algorithms [17,18] also suggests likewise.

The contributions of this paper are twofold. First, an effective clipping noise compensation scheme, consisting of a fully connected NN, is proposed. To the best of our knowledge, it is the first such NN used for this purpose. The simulation results demonstrate the superior performance of the proposed approach over existing methods, especially when clipping is severe. Second, the weight matrices of the trained NN are analyzed. Patterns resembling that of the DFT/IDFT matrices, which propagate information between the time and frequency domain, are revealed. It is rarely reported in DL literature that such recognizable patterns can be observed in NN weight matrices. Hopefully, this example can stimulate the emergence of more exciting research.

## 2. Clipping Noise Compensation

The proposed DL-based scheme seeks to reconstruct OFDM symbols in the frequency domain, such that the mean square error (MSE) is minimized. The overall schematic after frequency domain equalization is summarized in Figure 1. The signal model and the two major modules, namely the NN and the post-processing inspired by the Decision-Aided Reconstruction (DAR) algorithm [17], are also explained in this section.

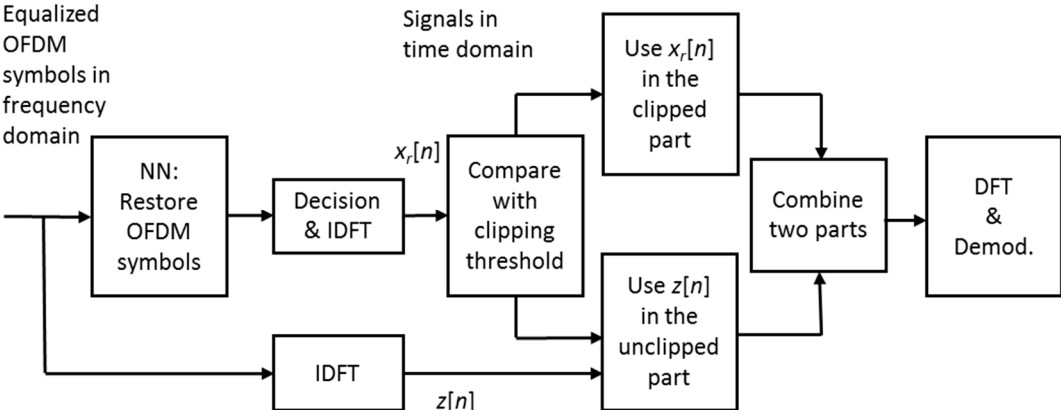

**Figure 1.** The schematic of reconstructing clipped orthogonal frequency division multiplexing (OFDM) signals.

Let $[X_0, X_1, \cdots, X_{N-1}]^T$ denote an OFDM symbol with $N$ subcarriers, where $X_k$ denotes the modulation (in this paper Quadrature Amplitude Modulation, QAM) symbol on the $k$-th subcarrier. The baseband signal in the time domain is

$$x[n] = \frac{1}{\sqrt{N}} \sum_{k=0}^{N-1} X_k e^{j2\pi nk/N}, \quad 0 \le n \le N-1. \tag{1}$$

To avoid inter-symbol interference (ISI) and to maintain the orthogonality between subcarriers, a short sequence called the cyclic prefix can be added to make the signal look "locally periodic" [1]. For instance, the last $L$ samples $x[N-L+1], \cdots, x[N-1]$ are added as prefix to make a complete OFDM symbol: $x[N-L+1], \cdots, x[N-1], x[0], x[1], \cdots, x[N-1]$. Usually, $L$ is chosen to be larger than the channel length to avoid ISI, and much less than $N$ to reduce overhead.

The digital clipping of OFDM signals is intentionally introduced to reduce the PAPR of the transmit signal. It is modeled as a restriction on magnitude, while the phase remains unchanged, that is

$$x_c[n] = \begin{cases} x[n], & \text{if } |x[n]| \leq A, \\ Ae^{j\theta}, & \text{if } |x[n]| > A, \end{cases} \tag{2}$$

where $x_c[n]$ denotes the clipped signal, $A$ is the clipping threshold and $\theta$ is the phase of $x[n]$. The clipping ratio is defined as

$$\text{cr} = \frac{A}{P_S} \text{ with } P_s = \text{E}\left[|x[n]|^2\right]. \tag{3}$$

Clipping is more severe when cr gets smaller. To achieve a PAPR reduction of 6 dB (5 dB), cr $\cong 1.0$ (1.3) is needed [6]. The signal is interpolated four times to emulate analog signals before doing digital clipping. Here, for the simplicity of presentation, the interpolation is not included.

At the receiver side, first the cyclic prefix is removed from the received OFDM symbol. As long as the length of the channel is no larger than $L$, the OFDM symbol will be immune to ISI and the orthogonality will also be maintained. Since the OFDM signal can be viewed as being circularly convoluted with the channel impulse response, it can be converted to frequency domain by DFT and then be equalized to compensate the channel gain. The equalization can be facilitated by piloted transmission, as is commonly done in communication protocols. To train the NN, the equalized symbols in frequency domain are used as the input and the correct OFDM symbols as labels. Assuming perfect Zero-Forcing (ZF) equalization, the equalized OFDM symbols are expressed as

$$Z_k = X_{c,k} + H_k^{-1} W_k \tag{4}$$

where $X_{c,k}$ is the clipped symbol which is the DFT of $x_c[n]$, $H_k$ the channel's frequency response, and $W_k$ the AWGN at the $k$-th subcarrier. Because the proposed scheme works on equalized OFDM symbols, it does not need to worry about compensating for channel gains, and we expect that the overall performance is also insensitive to channel conditions. Note that there exist extremely rich works on equalization and signal detection for OFDM systems. They range from the application of DL [9,10] to utilizing simple linear filters to combat complicated issues such as narrowband noise or inter-carrier interference [15,20–23]. It is possible that the proposed scheme will have a better performance if a more advanced equalizer other than the ZF equalizer is used.

Since the effect of clipping noise is spread over all subcarriers, a four-layer fully connected NN that exchanges information among all subcarriers is considered. The hyperbolic tangent (Tanh) is chosen as the activation function for its zero-centered property. The number of nodes is selected by trial and error to be three times the size of the input data. For example, there are 384 nodes in the hidden layer for an input data vector of 128 entries. MSE is used as the training criterion because the task can be viewed as a regression problem, and the ADAM optimizer [24] is applied with predefined learning rates (the initial value is 0.00006) for updating the weights. The depth of the NN is also determined by trials. In the end, four layers are used to present the BER performance. However, to present the interpretation of the weight matrices, a three-layer NN is used for a simpler presentation, though its BER performance is typically 0.1 dB worse than that of a four-layer NN.

The data are generated and split into training and testing sets. The input data are normalized to unit variance and the Xavier initialization scheme [25] is deployed to improve the convergence. Tensorflow [26] is used for the training sessions, but since it does not support complex number

operations, the complex-valued OFDM symbols are split into real parts and imaginary parts. The equalized OFDM symbol on the $k$-th subcarrier $Z_k$ is represented as

$$\widetilde{Z}_{2k} = \text{Re}(Z_k); \ \widetilde{Z}_{2k+1} = \text{Im}(Z_k). \tag{5}$$

The environment settings for the data generation are as follows: OFDM symbols with $N = 64$ subcarriers and a 16-tap cyclic prefix are used, and the modulation is 16 QAM since the clipping effect on four QAM is not prominent. To demonstrate that the proposed scheme is indeed insensitive to channel conditions, the training and testing data sets are generated with different channels. A six-tap Rayleigh fading channel (the channel length is less than the length of cyclic prefix $L = 16$) with an exponential power delay profile (PDP) is used for training the data, while an AWGN channel and a six-tap Rayleigh fading channel with uniform PDP are used for testing data. Finally, clipping ratios range from 1.0 (severe) to 1.3 dB (mild).

The hyperparameters of the NN model are as follows. The training set has 1.2 million data records, and four hundred thousand more are used for testing. The number of nodes in one hidden layer is 384. Approximately 1000 epochs, which denote the number of iterations to train through the entire dataset, are needed for the model to converge. The batch size, the number of data records used for training in one iteration, is 512.

The reconstructed signal by the NN, denoted as $\left[ X_{r,0}, \ \cdots, \ X_{r,k}, \cdots, X_{r,N-1} \right]^T$, is in the frequency domain. It can be directly passed on for QAM demodulation, or it can be refined further by a post-processing procedure inspired by the DAR algorithm [17]. The reconstructed signal is transformed and inspected in the time domain and two observations are made. First, the reconstruction can be improved further by cleaning up the frequency domain signal. This can be done by making a decision on $X_{r,k}$ before transforming it into the time domain [17]. That is, a better version of the time domain reconstructed signal, call it $x_r[n]$, can be obtained by

$$x_r[n] = \frac{1}{\sqrt{N}} \sum_{k=0}^{N-1} D\{X_{r,k}\} e^{j2\pi nk/N}, \quad 0 \le n \le N-1. \tag{6}$$

where $D\{\cdot\}$ means making a decision to the nearest QAM constellation point. The second observation is that the clipped part of the original $x[n]$ is reconstructed by $x_r[n]$ with great success. However, for the unclipped part, $x_r[n]$ can sometimes be distorted much away from $x[n]$. To cope with this issue, $x_r[n]$ is compared to the clipping threshold to establish the estimated range of the unclipped part. Then, the equalized signal $z[n]$, which is simply the IDFT of $Z_k$, will be used as the signal within the unclipped part, while $x_r[n]$ will be used as the reconstructed signal within the clipped part. The overall schematic is shown in Figure 1. Note that the clipping threshold can be obtained beforehand via communication protocols.

## 3. Results

In this section, the learned weight matrices of the NN are first presented, and patterns are recognized in them to provide a nice interpretation of their behavior. Then, the MSE of reconstructed signals and the final BER of the proposed scheme are provided. The performance is compared to that of the method based on Compressed Sensing (CS) [18] to illustrate the effectiveness of our approach. Our reason for choosing [18] is because it provides a state-of-the-art performance. There exist more up-to-date algorithms [27,28], which are modified from [18], to cope with more complicated nonlinear effects or to utilize more side information. They are not included here to keep the comparison straightforward.

### 3.1. Interpretation of NN Weight Matrices

Figure 2 visualizes the weight matrices of the learned three-layer fully connected NN. It is observed that these matrices exhibit interesting patterns. Some columns of the first weight matrix, denoted as $W_1$, have a single large entry, while other columns exhibit rhythmic fluctuations. These features must be reorganized into patterns that are easier to understand. Conventional clipping noise mitigation algorithms such as that in [17] require information in both time and frequency domain; this led us to focus on finding columns that are similar to those of an IDFT matrix (providing time domain information) and an identity matrix (preserving frequency domain information). Note that we also split the IDFT matrix into real and imaginary parts in a similar way to Equation (4). Assume $A$ is an $N \times N$ IDFT matrix, then the real-valued $2N \times 2N$ matrix $\hat{A}$ that can carry out complex number operations with $\widetilde{Z}_k$ is split into $\hat{A}_{2k,2k} = \mathrm{Re}(A_{k,k})$, $\hat{A}_{2k+1,2k} = -\mathrm{Im}(A_{k,k})$, $\hat{A}_{2k,2k+1} = \mathrm{Im}(A_{k,k})$ and $\hat{A}_{2k+1,2k+1} = \mathrm{Re}(A_{k,k})$.

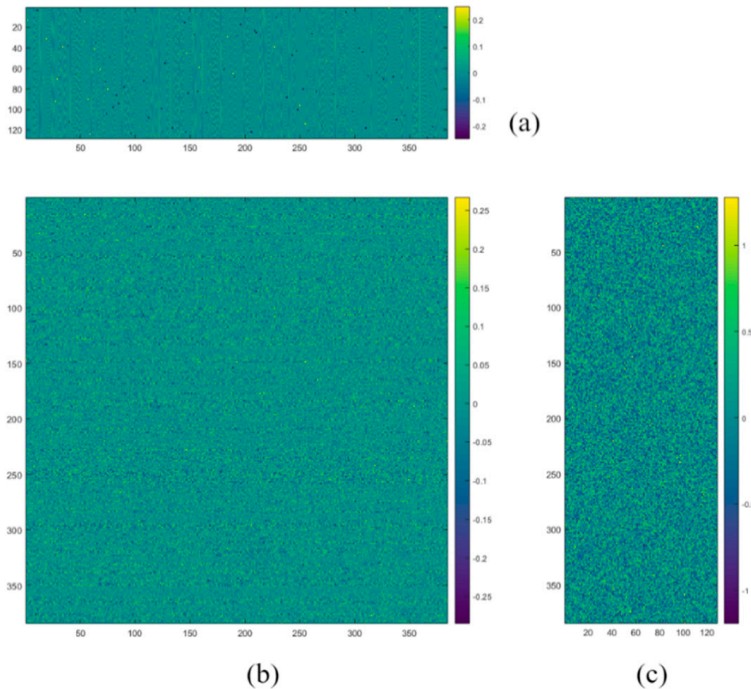

**Figure 2.** The weight matrices of the learned three-layer neural network. (**a**) $W_1$. (**b**) $W_2$. (**c**) $W_3$.

Figure 3b shows the $2N \times 2N$ IDFT matrix $\hat{A}$. The columns of $W_1$ that are similar (with high correlation) to the columns of the IDFT matrix and those similar to the columns of the identity matrix are both sorted out and reorganized. The selected columns can be reordered to make its similarity to the IDFT and the identity matrix more visible. The reordering operations can be recorded in a permutation matrix $E$, and the reordered weight matrix is denoted as:

$$\hat{W}_1 = W_1 E. \tag{7}$$

Figure 4a,b show the first and second layer weight matrix after permutation ($\hat{W}_1$ and $\hat{W}_2$). The left part of $\hat{W}_1$ is obviously similar to the identity matrix; the central part of $\hat{W}_1$ resembles the IDFT matrix and is singled out to be shown more clearly in Figure 3d; this suggests that the NN seeks to regenerate time domain information with frequency domain inputs. In visualizing the working of this NN, the nonlinear activation function is neglected to highlight the patterns emerging from the connections.

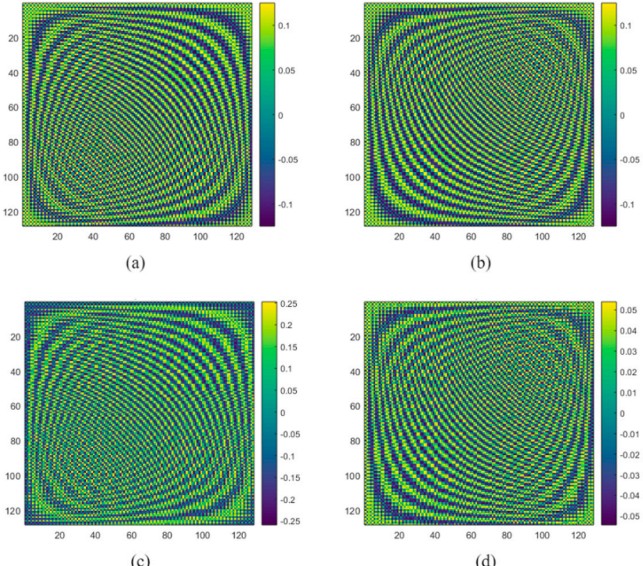

**Figure 3.** (**a**) DFT matrix, (**b**) IDFT matrix, (**c**) The learned DFT matrix (the central part of $\hat{W}_2 W_3$), (**d**) The learned IDFT matrix ($\hat{W}_1$).

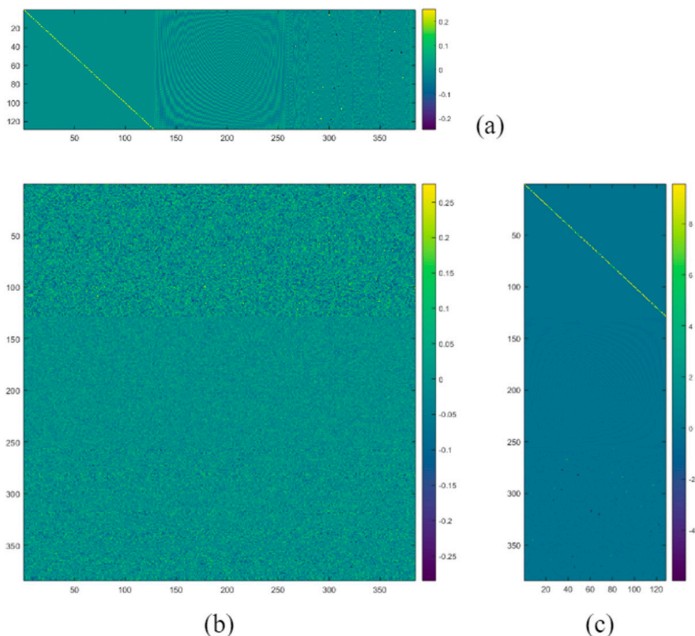

**Figure 4.** The reordered weight matrices. (**a**) $\hat{W}_1$ (**b**) $\hat{W}_2$ (**c**) $\hat{W}_2 W_3$.

The same permutation matrix is applied to the second layer weight matrix in order to preserve the connections between layers. On the premise that $EE^T = I$, we have

$$W_1 W_2 W_3 = W_1 \left(EE^T\right) W_2 W_3 = \hat{W}_1 \left(E^T W_2\right) W_3 = \hat{W}_1 \hat{W}_2 W_3. \tag{8}$$

It is not easy to see the patterns in $\hat{W}_2$ (shown in Figure 4b) and $W_3$ separately. However, when they are multiplied together, the resulting $\hat{W}_2 W_3$ again shows clearly recognizable patterns, this time resembling the identity and DFT matrices, as shown in the upper and central part of Figure 4c. The central part is, again, singled out in Figure 3c to be compared to the DFT matrix (Figure 3a). Since the output of the NN model is supposed to constitute a compensated OFDM symbol in the

frequency domain, it is no surprise that the third weight matrix tries to mimic the DFT matrix and convert data to the frequency domain.

The NN retains the original frequency domain input information from the learned identity matrix, while it mimics the behavior of IDFT and DFT. The IDFT is used to obtain the time domain information needed to estimate the clipping noise, while the DFT is needed to create compensated OFDM symbols in the frequency domain. We have tried to simultaneously use the time and frequency domain OFDM signals as the inputs to the NN, since the above interpretation suggests that they are both needed, yet, curiously enough, a better performance is not obtained from the training difficulties that are encountered due to the increased size of the NN.

### 3.2. MSE and BER Performances

In this section, two performance measures are presented: the MSE between the reconstructed and the original unclipped signal, and the BER of the un-coded OFDM system. For the MSE, two signal parts are evaluated separately, e.g., the clipped and unclipped part. The reason for doing this is to evaluate the capability of the NN to recover the clipped part while assess the potential damage that may occur to the unclipped part. The performance of the CS-based method [18] is compared.

Figure 5a,b show the MSE of the clipped parts with (a) clipping ratios = 1.0 and (b) 1.3. The blue dashed line indicates the MSE without clipping noise compensation. The other lines represent the MSE of various algorithms' MSEs. In general, the NN (yellow lines) outperforms the CS-based algorithm (orange lines), especially when the clipping is severe (cr = 1.0). The reason for this is because the necessary assumption of the CS-based algorithm that clipping occurs sparsely is violated when clipping is severe. The best performance happens when NN works with DAR-inspired post processing.

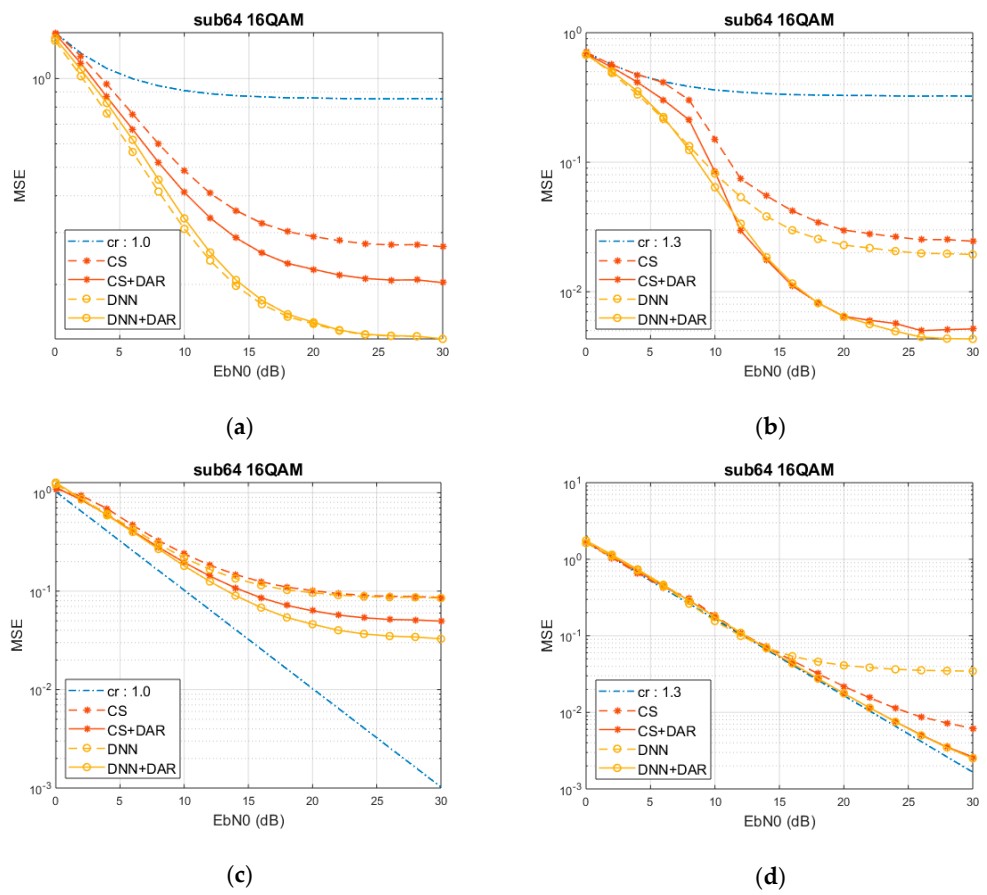

**Figure 5.** Mean square error (MSE) versus Signal-to-Noise Ratio (SNR): (**a**) clipped part, cr = 1.0 dB (**b**) clipped part, cr = 1.3 dB (**c**) unclipped part cr = 1.0 dB (**d**) unclipped part, cr = 1.3 dB.

Figure 5c,d show the MSE of unclipped parts. The DL-based algorithm suffers from signal distortion caused by the NN and has the worst MSE when the protection of the unclipped part is not activated (Figure 5d). DAR-inspired post processing, as described in Section 2, can be applied to improve the performance of the DL-based algorithm. As a result, the MSE performance is greatly improved. Notice that the protection can be applied to the CS-based algorithm as well. Overall, taking into account the MSE of both the clipped and unclipped parts, the DL-based scheme achieved a better performance.

Next the BER performance is presented under various clipping ratios and channel conditions. Only two NN models are needed to cover the wide range of clipping ratios. They are trained with the clipping ratio 1.08 and 1.27, and neighboring clipping ratios can be covered by them. However, note that the design of the NN is not scalable yet. The NN model can only work for a fixed number of subcarriers.

In Figure 6a,b, with AWGN and Rayleigh fading channels, respectively, the dashed lines and solid lines represent the BERs with and without compensation by the DL-based method, and the improvement is significant. Figure 6c,d, with AWGN and Rayleigh fading channels, respectively, the BERs of the CS- and the DL-based method are compared; again, the DL-based method outperforms the CS-based algorithm by a large margin.

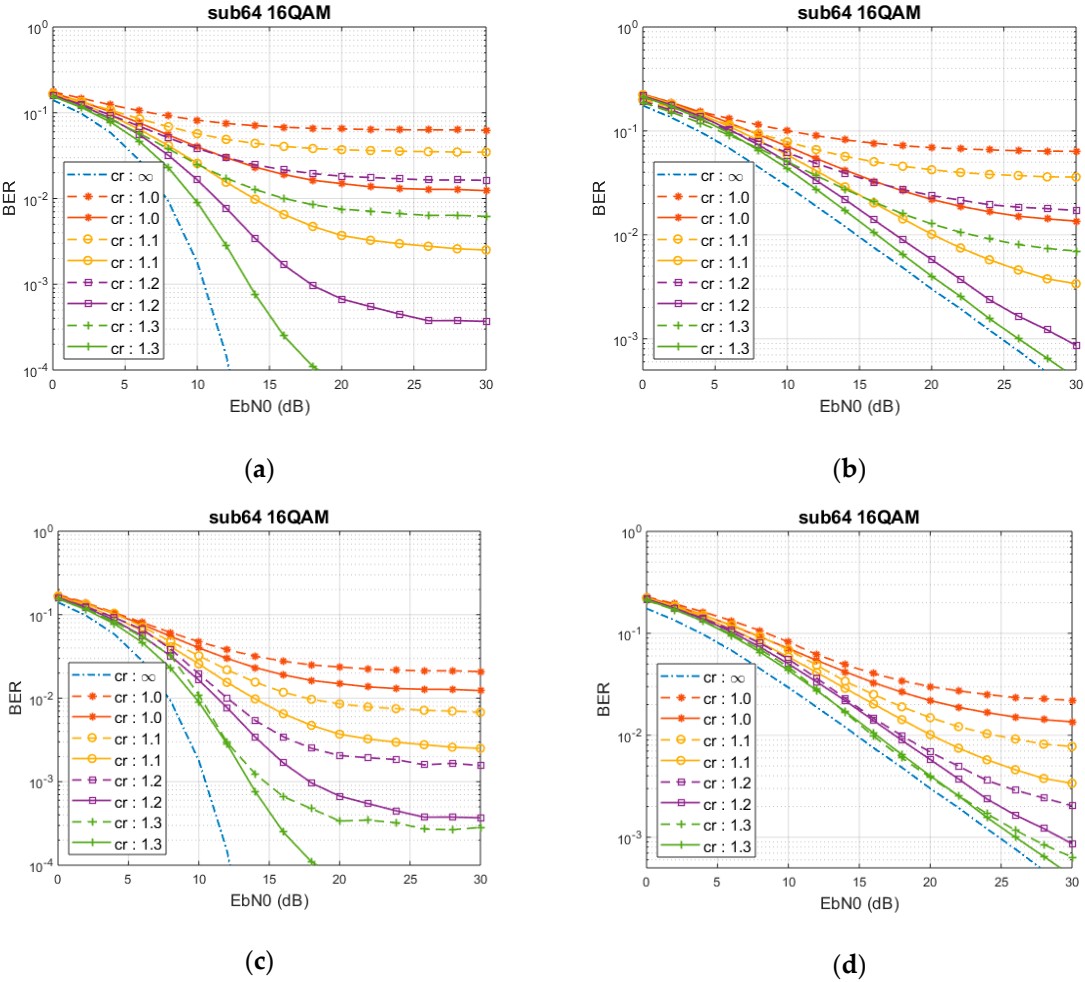

**Figure 6.** BER performance in AWGN and Rayleigh fading channels with different clipping ratios. In (**a**,**b**), dashed line: no clipping noise compensation; solid line: Neural Network (NN). In (**c**,**d**), dashed lines: CS; solid lines: NN.

## 4. Conclusions

In this article, a DL-based scheme to mitigate the clipping effects in OFDM systems is demonstrated. A four-layer NN is used and its output is post processed with a DAR-inspired post processing to protect the unclipped part from undesired distortions. The simulation results show that the MSE and BER performances under different clipping ratios and channel conditions are superior to that of the existing algorithms. To interpret the behavior of the NN, the hidden patterns of the trained weight matrices are revealed, and they facilitate our understanding of the neural network.

**Author Contributions:** The contributions to this article include: conceptualization, T.-H.S. and Y.-C.X.; methodology, T.-H.S. and Y.-C.X.; software, Y.-C.X.; formal analysis, T.-H.S. and Y.-C.X.; investigation, T.-H.S. and Y.-C.X.; resources, T.-H.S.; data curation, Y.-C.X.; writing—original draft preparation, Y.-C.X.; writing—review and editing, T.-H.S.; visualization, Y.-C.X.; supervision, T.-H.S.; project administration, T.-H.S.; funding acquisition, T.-H.S. All authors have read and agreed to the published version of the manuscript.

**Funding:** This research was partially funded by the "Center for mmWave Smart Radar Systems and Technologies" under the Featured Areas Research Center Program within the framework of the Higher Education Sprout Project by the Ministry of Education (MOE), and partially funded by the Ministry of Science and Technology (MOST) of Taiwan under MOST 108-3017-F-009-001 and MOST 108-2218-E-009-018.

**Acknowledgments:** This work was partially supported by the "Center for mmWave Smart Radar Systems and Technologies" under the Featured Areas Research Center Program within the framework of the Higher Education Sprout Project by the Ministry of Education (MOE), and partially supported under MOST 108-3017-F-009-001 and MOST 108-2218-E-009-018, in Taiwan.

**Conflicts of Interest:** The authors declare no conflict of interest.

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
