# Peer review of "Clipping Noise Compensation with Neural Networks in OFDM Systems"

_signals, 2013_

Round 1
Reviewer 1 Report
The paper "Clipping noise compensation with neural networks in OFDM systems" deals with an interesting topic. The paper is generally clear. However, the Authors should explain better the signal model (3). It's true that OFDM is a well-known technology, but the Authors have to specify the hypotheses under which this model holds. In particular, they have to mention the cyclic prefix (CP) insertion, they have to present the channel model and give further detail on its memory Lh with respect to the CP length, because only if Lh < Lcp you can write eq. (3) (in this case CP removal at the receiver is effective). The Reviewer suggests the following papers:
[R1]: "Widely linear equalization and blind channel identification for interference-contaminated multicarrier systems", D Darsena, G Gelli, L Paura, F Verde IEEE Transactions on Signal Processing 53 (3), 1163-1177
[R2]: "A constrained maximum-SINR NBI-resistant receiver for OFDM systems", D Darsena, G Gelli, L Paura, F Verde, IEEE transactions on signal processing 55 (6), 3032-3047
[R3]: "Impulse noise mitigation for MIMO-OFDM wireless networks with linear equalization", D Darsena, G Gelli, F Melito, F Verde, A Vitiello, 2013 IEEE International Workshop on Measurements & Networking (M&N), 94-99
The Authors should take into account what has been done in the literature on this topic. Otherwise the application of ML to communication problems become a pure exercise.
Some minor remarks:
1) some typos should be fixed (e.g., "which of" in the Abstract 3rd line);
2) it would be better to indicate the ratio cr in math mode (as A and Ps).
Reviewer 2 Report
PAPR reduction in OFDM systems always is an interesting topic; digital clipping is the simplest, and mainly used approach for mitigating PAPR. The authors propose a clipping reduction algorithm based on deep learning framework.
Authors are simply using consolidated neural network algorithms to mitigate the effects of the clipping noise. Nothing new on the used tools, the novelty should consist on the application to the problem.
The clipping noise effects can be overcome using particular filter implementations (e.g., MMSE and non-local mean), but the authors only consider a compressive sensing approach. Please consider other techniques for comparison.
The understanding of the paper is limited by the language, I noticed a lot of mistakes that can distract the reader. Please have an insightful review.
Round 2
Reviewer 1 Report
The Authors have taken into account my concerns. The paper is ready for publication.
Reviewer 2 Report
All of the suggested modifications have been done.
Good work.